# A Preliminary Study of Robust Speech Feature Extraction Based on Maximizing the Probability of States in Deep Acoustic Models

**Li-Chia Chang and Jeih-Weih Hung** *

Department of Electrical Engineering, National Chi Nan University, Nantou 545, Taiwan;
s108323518@mail1.ncnu.edu.tw
* Correspondence: jwhung@mail.ncnu.edu.tw

**Abstract:** This study proposes a novel robust speech feature extraction technique to improve speech recognition performance in noisy environments. This novel method exploits the information provided by the original acoustic model in the automatic speech recognition (ASR) system to learn a deep neural network that converts the original speech features. This deep neural network is trained to maximize the posterior accuracy of the state sequences of acoustic models with respect to the speech feature sequences. Compared with the robustness methods that retrain or adapt acoustic models, the new method has the advantages of less computation load and faster training. In the experiments conducted on the medium-vocabulary TIMIT database and task, the presented method provides lower word error rates than the unprocessed baseline and speech-enhancement-based techniques. These results indicate that the presented method is promising and worth further developing.

**Keywords:** speech recognition; noise-robust acoustic features; deep learning





## 1. Introduction

Electronic devices and services such as mobile phones, hearing aids, headphones, and teleconference systems play a significant role in our lives. In particular, voice-based functions and applications (voice interaction, voice communication, and speech recognition) are essential in these devices and services. However, various sources of interference exist that deteriorate speech signals during transmission, and thus they undermine the capability of the functions mentioned above and applications. These interference sources include additive noise, channel distortion, and reverberation. In order to alleviate the influence of interference, various methods have been presented in recent decades from different perspectives of speech processing systems, such as front-end signal processing, acoustic feature design, and back-end acoustic modeling. Roughly speaking, the methods regarding the front-end acoustic signal-domain processing belong to speech enhancement (SE). In contrast, the techniques for the acoustic features and models are related to robust speech recognition.

According to [1], speech enhancement (SE) algorithms can be divided into four categories: spectral-subtractive [1–3], statistical modeling [1,4–7], sub-space based [1,8,9], and masking-based [10–12]. In particular, due to the successful development of deep neural network (DNN) techniques in the recent decade, the modeling of speech enhancement algorithms and the respective procedures (such as noise estimation and sub-space basis extraction) are significantly upgraded to achieve even better performance. For example, a DNN can be used to learn the statistical relationship between noisy utterances and clean counterparts through a massive amount of noisy–clean utterance pairs in a prepared training set. In particular, since speech enhancement aims to transform noisy speech back to clean speech, which is a standard regression problem in machine/deep learning, the mean squared error (MSE) loss function is often used in the SE-wise DNN.

However, when it comes to the performance evaluation of a speech enhancement algorithm, some objective metrics, such as the perceptual evaluation of speech quality (PESQ) [13] and short-time objective intelligibility (STOI) [14], are often used. These metrics are not necessarily related to the mean squared distance between the enhanced and original clean speech. Therefore, in some recently developed deep-learning-based SE algorithms [15,16], PESQ and STOI are employed directly to be the objective functions for the training of DNN models.

PESQ is a standard objective metric to measure speech quality, recommended by ITU-T. It is developed to predict the mean opinion scores (MOS) in subjective listening tests, and according to [17], PESQ shows a high correlation ($r \approx 0.9$) with MOS. PESQ is widely used to evaluate speech separation and enhancement algorithms. It mainly quantizes speech quality by computing the disturbance between the clean and separated/enhanced speech using cognitive modeling. PESQ ranges within $[-0.5, 4.5]$, with high values indicating better quality [13,18].

Here, we provide a little more discussion about the issue that using mean squared error (MSE) as the loss function in most deep-learning-based SE algorithms [10,19–21] does not necessarily give rise to higher PESQ/STOI scores in the processed utterances. According to [18,21], two fundamental assumptions of the MSE loss in these SE methods somewhat contradict human auditory perception, which are as follows:

- **Every estimated element is equally important [18]:**
  The MSE loss function treats each single data point equally. Regardless of the locations of the processed data, the same difference between the elements of each processed output and the desired output corresponds to the equal MSE loss value. The underlying reason is that the MSE loss is derived from each sample point of the utterance separately and independently instead of the whole utterance trajectory. However, we evaluate the quality and intelligibility of an utterance in a different manner. For example, the speech-dominant portion is treated more crucially than the silence portion in the STOI metric.
  In contrast, the PESQ metric pays more attention to isolated samples because they are more involved with speech quality. In addition, when considered in the frequency domain, the MSE loss is usually defined at the linear frequency scale. At the same time, the speech quality/intelligibility metrics are measured at human auditory frequency scales (such as the mel scale and Bark scale). To sum up, minimizing the MSE loss is not necessary to optimize the PESQ or STOI scores for noisy utterances.

- **The optimal solution is determined by the whole utterance set [21]:**
  To extend the idea of the previous point, using the MSE as the objective loss function tends to consider the portions in utterances with different characteristics in an *averaged* manner, which might alleviate the distinct and discriminative components that highlight the speech. For example, an ordinary utterance set possesses low- and high-pitch utterances. Minimizing the MSE at these utterances indiscriminately very likely causes these utterances to have median pitches, which is not our goal at all.

Inspired by the observations mentioned above and some other literature [15,18,22], in this study, we propose a novel deep-learning-based noise-robust speech feature extraction algorithm with an MSE-irrelevant loss function. The used loss function is directly associated with the performance of a speech recognition system in noisy environments. Briefly speaking, a deep neural network is trained to perform the speech feature mapping that maximizes the posterior state probability associated with the back-end acoustic models in the speech recognition system. The resulting new speech feature representation is expected to outperform the original feature in recognition accuracy and possess noise robustness.

It is noteworthy that many research efforts have employed deep neural networks to engage in an ASR system to improve its noise robustness. For example, an end-to-end (E2E) ASR framework with a single DNN directly maps an acoustic feature sequence to a word sequence [23–25]. This E2E DNN is learned to optimize criteria directly associated with the ASR performance, such as the word error rate (WER). Such an E2E model can

behave better and be more compact than the conventional hybrid ASR model, consisting mainly of acoustic models, a lexicon, and language models. The authors in [26] claim that the improvements in speech quality metrics such as PESQ by speech enhancement techniques do not translate into better ASR performance. They propose a joint optimization of mask-estimating network-based speech enhancement and acoustic modeling to reduce the WER. In addition, a deep learning scheme for *a priori* SNR estimation, termed Deep Xi, is presented in [27,28] to facilitate the conventional minimum-mean-squared-error (MMSE)-based SE methods for a robust ASR. Compared with these complicated and fine-grained techniques, our newly presented method is a relatively lightweight network that is much easier to be learned, while it is likely less effective. However, our method is a DNN-wise transformation in the same feature domain and thus can be easily integrated with these advanced methods to boost the ASR performance.

In the following sections, we first introduce the original acoustic features and the associated acoustic models in a pre-processing stage and then present the novel feature extraction algorithm. Next, we compare our method with various speech features created by the utterances enhanced by some state-of-the-art enhancement methods. Finally, we give a concluding remark.

## 2. Introduction of Speech Feature Representation and Acoustic Model Configuration

In this section, we set forth the process of creating the widely used speech feature representation: mel-frequency filter-bank features (FBANK) and mel-frequency cepstral coefficients (MFCC). After that, we introduce the configuration of a deep-neural-network-based hidden Markov model (DNN-HMM), which is widely used as the acoustic model structure in the current automatic speech recognition (ASR) system. This section serves as the background knowledge for a novel noise-robust speech feature extraction algorithm, which is clarified in the next section.

### 2.1. Mel-Frequency Cepstral Coefficients and Mel-Frequency Filter-Bank Features

Mel-frequency filter-bank features (FBANK) and mel-frequency cepstral coefficients (MFCC) are two of the most widely used speech feature representations in the state-of-the-art ASR system. FBANK and MFCC approximate the response of the human auditory system to sounds better than the linearly scaled frequency-band energies (short-time Fourier transforms of sound segments), thereby corresponding to superior recognition accuracies in ASR.

The procedures of creating a series of FBANK and MFCC features from an utterance are depicted in Figure 1. From this figure, it is clear that computing FBANK is the same as computing MFCC except for the discrete cosine transform (DCT) in the final step. The DCT operation is believed to reduce the correlation of the input significantly, and thus MFCC features are much less correlated than FBANK features. It is noteworthy that in conventional ASR systems, MFCC is preferred over FBANK due to its nearly uncorrelated characteristic. However, FBANK usually behaves better than MFCC in some advanced ASR systems whose acoustic models are non-sensitive to feature correlation.

In the following, we briefly introduce each MFCC/FBANK creation procedure shown in Figure 1.

1.  *Pre-emphasis:*
    The first step is to apply a pre-emphasis filter on the input signal to amplify the high-frequency components because utterances usually have smaller magnitudes in high frequencies compared to low frequencies. The pre-emphasis filter can be applied to a signal $x[n]$ as the following equation:

    $$y[n] = x[n] - 0.97x[n-1]. \tag{1}$$

2. *Framing:*
An utterance is non-stationary, and its inherent characteristics change over time. However, it is reasonable to assume that the characteristics remain unchanged in each short-time interval. Thus, the utterance is segmented into a series of partially overlapped frames. The frame duration and frame shift are usually set to 25 ms and 10 ms, respectively.

3. *Windowing:*
After framing, each frame signal is multiplied with an appropriate window function to reduce the unwanted sidelobes in spectrum [29]. We usually use the Hamming window function here, which is:

$$w[n] = 0.53836 - 0.46164 \cos(\frac{2\pi n}{N-1}), n = 0, 1, \ldots, N-1, \tag{2}$$

where $N$ is the frame size.

4. *Discrete Fourier transform (DFT):*
The windowed time-domain frame signal is converted to the acoustic frequency domain via a discrete Fourier transform (DFT). In the frequency domain, the signal's human hearing characteristics (such as formants and pitches) can be better revealed. The function of the DFT is as follows:

$$X[k] = \sum_{n=0}^{N-1} x[n] e^{-j\frac{2\pi kn}{N}}, k = 0, 1, \ldots, N-1. \tag{3}$$

5. *Mel-frequency wrapping:*
The perceived frequency resolution in the human hearing mechanism decreases as the physical frequencies increase, making humans less sensitive to frequency change at high frequencies. The mel-filter-banks, which consist of triangular band-pass filters, are designed to simulate the above mechanism by locating more narrow-bandwidth filters at low frequencies and fewer wide-bandwidth filters at high frequencies. The $l$th mel-filter $H_l[k]$ of the filter-bank is defined by:

$$H_l[k] = \begin{cases} 0, & k < f[l-1] \\ \frac{k-f[l-1]}{f[l]-f[l-1]}, & f[l-1] \le k \le f[l] \\ \frac{f[l-1]-k}{f[l-1]-f[l]}, & f[l] \le k \le f[l+1] \\ 0, & k > f[l+1] \end{cases} \tag{4}$$

in which $f[l]$ denotes the center frequency of the $l$th mel-filter, calculated by:

$$f[l] = \frac{N}{f_s} mel^{-1}[mel(f_l) + l\frac{mel(f_H) - mel(f_L)}{J+1}] \tag{5}$$

with $f_H$ and $f_L$ being, respectively, the highest and lowest frequencies and

$$mel(f) = 2595 \log_{10}(1 + \frac{f}{700}) \tag{6}$$

which transforms the acoustic frequency $f$ (in Hz) to the perceptual frequency $mel$. With the set of mel-filters, we convert the magnitude spectrum $|X[k]|$ for each frame to the mel-scaled (magnitude) spectrum:

$$S[l] = \sum_{k=0}^{\frac{N}{2}-1} H_l[k] |X[k]|, l = 1, 2, \ldots, L \tag{7}$$

6. *Logarithmic operation:*
The human auditory system tends to adjust the strength of the received sound to protect the ears. When the input sound has a high amplitude, it will be suppressed by

the human ears. To portray such a function, a logarithmic operation is applied to the mel-scaled spectrum $S[l]$ as follows:

$$S^l[l] = \log(S[l]), l = 1, 2, \ldots, L \tag{8}$$

In particular, the features $S^{(l)}[l]$ obtained in Equation (8) are called logarithmic mel-scale filter-bank spectra, which are often referred to as FBANK features.

7. *Discrete Cosine Transform (DCT):*
   The FBANK features can be converted to the cepstral domain via the discrete cosine transform (DCT). As mentioned earlier, the DCT has the ability to produce nearly uncorrelated features, and here the resulting outputs are MFCC features. The DCT operation to obtain the MFCC, $y[p]$ from FBANK $S^{(l)}[l]$, is as follows:

$$y[p] = \frac{1}{L} \sum_{l=1}^{L} S^{(l)}[l] \cos\left(\frac{\pi p(l - 0.5)}{L}\right), p = 0, 1, \ldots, J - 1 < L \tag{9}$$

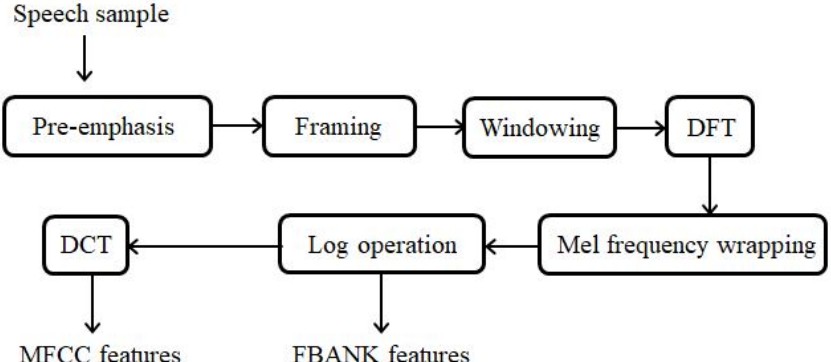

**Figure 1.** Flow chart of MFCC calculation.

*2.2. Hidden Markov Model with Deep Neural Networks*

In an ASR system, while observing a sequence of acoustic features $O = [o_1, o_2, o_3, \ldots, o_T]$ (e.g., FBANK or MFCC), the objective is to find the most likely word sequences $W^*$ among all possible word sequences $W = (w_1, w_2, w_3, \ldots, w_T)$. That is,

$$W^* = \arg\max_W P(W|O) = \arg\max_W P(O|W)P(W), \tag{10}$$

where $P(W|O)$ is the posterior probability of $W$, which is proportional to $P(O|W)$ and $P(W)$ computed by the acoustic models and language models, respectively.

Among the various acoustic models, a hidden Markov model (HMM) is a double stochastic process consisting of states, each describing the observations (speech features). An HMM is parameterized by state-transition probability and state-dependent observation probability. In particular, the state-dependent observation probability has been usually computed via a Gaussian mixture model (GMM) in the past. However, it is more frequently evaluated with a deep neural network (DNN) in the state-of-the-art ASR. In the following, we briefly describe the DNN-HMM [30].

1. *State-dependent observation probability [31]:*
   An HMM acoustic model consists of several states, and each state corresponds to a probability function that describes the random observation vector. To train a deep neural network (DNN) that exhibits the probability of observations for each state, a common procedure is to train a conventional GMM-HMM (using a GMM to describe each state) at first and then use the GMM-HMM to provide each observation (feature) in a training set with a state label. After that, supervised learning is conducted on the observations as the input together with the state labels as the desired output to obtain the deep neural networks (DNNs) for the states of the HMM. We often use the

softmax activation for the output layer of the DNN and the log-likelihood as the loss function, and thus the resulting DNN can output the state-dependent observation probability, denoted by $b_j(o_t)$, where $j$ is the state index and $o_t$ is the observation at an arbitrary time index $t$.

2. *State-transition probability:*
   Each DNN-HMM acoustic model usually corresponds to a single mono-phone or a tri-phone. To find the most likely state sequence in an HMM for an arbitrary input utterance, the Viterbi algorithm is often employed. Let $\lambda$ refer to the model parameters of a specific HMM, and $O = [o_1, o_2, o_3, \ldots, o_T]$ is the observation sequence corresponding to that HMM. The Viterbi algorithm tries to find the probability:

$$\delta_t(i) = \max_{q_1 q_2 \ldots q_{t-1}} [P(q_1 q_2 \ldots q_{t-1}, q_t = i, o_1 o_2 \ldots o_t | \lambda)] \tag{11}$$

for $t = 1, 2, \ldots, T$ in series and finally find the best state sequence that produces the highest probability $P^* = \max_{1 \leq i \leq N} [\delta_T(i)]$. It can be found that:

$$\delta_{t+1}(j) = \max_i [\delta_t(i) a_{ij}] b_j(o_{t+1}), \tag{12}$$

where $a_{ij}$ is the transition probability from the $i$th state to the $j$th state. According to Equation (12), $\delta_{t+1}(j)$ considers both the observation probability $b_j(o_{t+1})$ and state-transition probability $a_{ij}$ of the HMM model.

## 3. The Presented New Method for Noise-Robust Feature Extraction

We propose a novel method to create noise-robust speech features in ASR. This method enhances the original acoustic features in noise robustness for an ASR system without alternating the acoustic models. In the following, we introduce the proposed method and then discuss its characteristics.

### 3.1. The Procedures of the Presented Method

In Section 1, we discuss the issue of using the mean squared error (MSE) as the loss function to train the speech enhancement model. The resulting model does not necessarily benefit speech quality and intelligibility in terms of the PESQ and STOI evaluation scores. The underlying reason is the objective mismatch, i.e., minimizing the MSE loss function versus maximizing the evaluation scores, PESQ and STOI. Inspired by this observation, in this study, we present a training algorithm for noise-robust speech feature extraction in which the loss function is directly related to the recognition accuracy. Briefly speaking, the main idea of our proposed method is to find the acoustic speech features $o_t'$ in noisy environments that maximize the accuracy of state sequences $q$ in HMMs with respect to the clean noise-free condition. The flowchart of the presented method is shown in Figure 2, and it consists of the following steps:

**Step 1:** We compute the FBANK and MFCC features for every utterance in the training set. Then, the features are further processed by mean and variance normalization (MVN), resulting in the final features $o_t$, where $t$ is the frame index.

**Step 2:** With the MFCC features in the training set, the GMM-HMM acoustic models (mono-phones and tri-phones) are trained following the standard recipes of Kaldi [32]. Moreover, linear discriminant analysis (LDA) [33], maximum likelihood linear transform (MLLT) [34], and speaker-adaptive training (SAT) [35] are applied to the speech features during the model training. After training, we obtain the GMM-HMM aligned state sequences $\bar{q}$ for each feature sequence $o_t$ in the training set, and the state sequence includes mono-phone labels and tri-phone labels.

**Step 3:** With the FBANK features and their state labels for the training set in hand, we train the corresponding DNN-HMM acoustic models [31]. It is a multi-classification task since we aim to find the transformation between the speech features $o_t$ and

their corresponding state labels $q_t$. The final layer of each DNN can produce the state-dependent observation probability of each $o_t$. It is worth noting that here we adopt multi-condition speech training set in the DNN learning stage. As such, the resulting DNN-HMMs are expected to be more noise-robust than the original GMM-HMMs. With the more accurate state sequences for each frame, combined with the trained HMM model in Step 2, we could obtain the word error rates (WER, %) which are the evaluation scores in our experiment.

**Step 4:** This step is the core of our presented method. Here, we train a denoising deep neural network to transform the original speech feature $o_t$ to another representation $o'_t$ as follows:

$$o'_t = f_{DN}(o_t), \tag{13}$$

where $f_{DN}(.)$ denotes the mapping function of the trained network. The objective of the trained network is to make the resulting new features $o'_t$ predict a state sequence from the DNN-HMM created in Step 3 that is closest to the ground-truth state sequence. We express the underlying idea in the following:

$$f_{DN} = \arg\max_f (Acc(q, q' = g(f(o_t)|\lambda))) \tag{14}$$

where $\lambda$ is the DNN-HMM acoustic model, $g$ is the function to produce the highest-likelihood state sequence, $q'$, of new features $f(o_t)$ given the model $\lambda$, $q$ is the ground-truth state sequence, and $Acc$ is the log-likelihood function to measure the accuracy of $q'$ relative to $q$.

Once the denoising deep neural network $f_{DN}$ is trained, in the recognition process we use it to map the speech features $o_t$ in a noise-corrupted utterance to new features $o'_t$. Then, we feed $o'_t$ to the *original* (not re-trained) DNN-HMM acoustic models together with language models in order to produce the maximum-likelihood state sequence and the word sequence. Compared with the original features $o_t$, the new features $o'_t$ are supposed to be more noise-robust since they are created with the help of the multi-condition DNN-HMMs. In addition, the new features $o'_t$ are supposed to produce lower word error rates in recognition since they correspond to the state sequence with a higher state posterior accuracy relative to the original $o_t$.

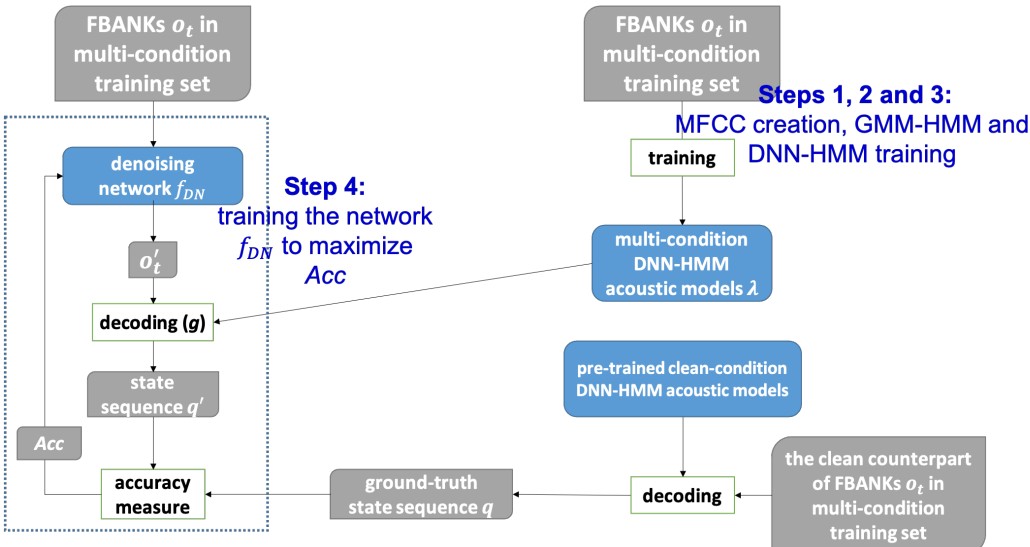

**Figure 2.** The flowchart of proposed method.

### 3.2. The Characteristics of the Proposed Method

Compared with some DNN-based speech feature extraction methods using MSE as the loss function [36], the newly presented method has the following potential advantages:

1.  Our method aims to maximize the state posterior accuracies of multi-condition acoustic models given the noise-corrupted data, which implicitly reduces the word error rates (WERs) for noisy speech recognition. In contrast, those methods minimizing the MSE among the clean–noisy speech feature pairs in the training set have little to do with the subsequent acoustic models and thus are not guaranteed to improve the recognition accuracy for noisy situations.
2.  Our method employs the multi-condition training set to obtain GMM-HMM and DNN-HMM acoustic models in turn, which serve to train the denoising feature extraction network. However, we expect such a denoising framework to work well when using clean noise-free training data. The underlying reason is that the framework adapts the original speech features to fit the subsequent acoustic models. In the later section, we observe and discuss the corresponding results from the evaluation experiments.

## 4. The Evaluation Results and Discussions

This section provides a series of experiments to evaluate the presented new noise-robust speech feature extraction method. The experimental setup is given in Section 4.1, and the recognition results achieved by some well-known speech enhancement methods are given and discussed in Section 4.2. Finally, Sections 4.3 and 4.4 present and analyze the presented method's results as well as several comparative algorithms.

### 4.1. The Experimental Setup

In the following, we describe the used data corpus, the ASR system preparation, and the arrangement of the DNN for our presented method.

#### 4.1.1. Data Corpus

We used the well-known TIMIT corpus [37] to experiment. TIMIT contains phonemically and lexically transcribed speech utterances from American English speakers of different genders and dialects. These utterances correspond to the broadband recordings of 630 speakers of eight major dialects of American English, each speaker reading ten phonetically rich sentences. Each utterance is a 16-bit, 16 kHz sampling rate waveform file accompanied by time-aligned orthographic, phonetic, and word transcriptions. The utterances in TIMIT were used as the training and test sets in our evaluation experiments. As for the multi-condition training set, 1000 utterances were randomly selected and corrupted with any of three noise types (babble, car, and street) at five **signal-to-noise ratio (SNR)** levels ($-5$ dB, 0 dB, 5 dB, 10 dB, and 15 dB). For the test set, 400 utterances different from the training set were selected and corrupted with any of three noise types (white, engine, and jackhammer) at six SNR levels ($-6$ dB, $-3$ dB, 0 dB, 3 dB, 6 dB, and 12 dB). In the experiments, we tested the utterances with respect to each noise type and each SNR level to see the detailed effects of the evaluated methods.

#### 4.1.2. The ASR System Preparation

The speech features in the training set were used to learn context-dependent (CD) tri-phones as acoustic models, which were arranged to be two different structures, i.e., GMM-HMM and DNN-HMM, respectively. Stating in more detail, GMM-HMM and DNN-HMM used GMM and DNN, respectively, to represent each state of the HMM. As for GMM-HMM, each mono-phone of speech signals and the silence were respectively characterized by an HMM with three states, having 1000 Gaussians in total, and each tri-phone was characterized by an HMM with three states, having 2500 leaves and 1500 Gaussians in total. Furthermore, LDA, MLLT, SAT were applied to the speech features during the tri-phone model training. On the other hand, for the DNN structure, five hidden layers, each containing 1024 nodes with a dropout rate 15%, were connected to

two independent output layers for tri-phones and mono-phones, respectively, in the DNN-HMM. The training process of this DNN underwent 24 epochs with an SGD optimizer and used the log-likelihood loss function for tri-phones and mono-phones. The losses of tri-phones and mono-phones were added up for minimization in the model training. The Kaldi toolkit [32] was used to create the GMM-HMM and the Pytorch-Kaldi [38] toolkit was used to create the DNN-HMM.

In addition, a set of tri-gram language models for training utterances was created via following the standard recipes of Kaldi.

### 4.1.3. The DNN Structure for Denoising Feature Extraction

For each of the utterances in the training and test set, the 69-dimensional FBANK feature stream (23 static FBANKs together with their delta and delta-delta for each frame, with a 20 ms frame duration and a 10 ms frame shift) was created as the baseline feature representation. The presented denoising DNN framework took the FBANK as input to produce the updated feature following the procedures in Section 3.1 for the subsequent recognition. The denoising DNN model was a convolutional neural network (CNN) with four same-size one-dimensional convolutional layers, each following a configuration setting (30, 5, 2), where the setting representation was "(number of kernel, kernel size, number of padding)". In addition, these four convolutional layers were followed by the two same fully connected layers with 759 nodes. The activation function for each layer output was the rectified linear unit (ReLU). The training process of this denoising framework underwent 30 epochs with the Adam optimizer and used the log-likelihood loss function.

### 4.2. The Effect of SE Methods in Speech Recognition for Noisy Speech

In this subsection, we would like to investigate whether the signals pre-enhanced by speech enhancement (SE) methods can result in speech features that bring about improved recognition accuracies relative to the baseline. We selected two SE methods: MMSE [39] (an unsupervised approach) and ideal ratio mask (IRM) [10] (a supervised approach) and conducted the evaluation experiments with a multi-condition mode in two different arrangements:

(1)   The SE process was conducted on the utterances in both the training and test sets.
(2)   The SE process was only conducted on the utterances in the test set, while the training set remained unchanged.

Tables 1–3 list the word error rates (WER in %) for the above two arrangements with respect to the test utterances of three noise types (white, engine, and jackhammer). In these tables, MMSE (TR, TS) and IRM (TR, TS) refer to the first arrangement, in which both the training and test sets were enhanced by MMSE/IRM, while MMSE (TS) and IRM (TS) refer to the second arrangement, in which MMSE/IRM enhanced only the test set. Furthermore, to see if the two SE methods can improve the quality and intelligibility of the test utterances as they promise, we also list the PESQ and STOI scores in these tables. From the results shown in these three tables, we have the following observations:

1.   Both MMSE and IRM improve the PESQ scores, while MMSE behaves better than IRM in most cases. For example, MMSE obtains 1.647 and 1.967 in PESQ at the 12 dB SNR case of Tables 1 and 2, while the corresponding values achieved by IRM are 1.275 and 1.493. This is possibly due to the mismatch between the training and test sets for IRM. However, neither MMSE nor IRM can bring a significant STOI upturn for almost all noise scenarios. In addition, IRM results in less STOI degradation than MMSE.
2.   Comparing the two first arrangements ("TR, TS" and "TS") as for MMSE and IRM, we find that conducting the SE methods on both the training and test sets shows inferior recognition performance (higher WERs) than conducting them only on the test set. Furthermore, the first arrangement "TR, TS" even causes worse results than the baseline. For example, the methods "MMSE (TR, TS)" and "MMSE (TS)" obtain 56.5% and 55.0% WERs at the 3 dB SNR case of Table 1, which are worse than the baseline 49.8%. The underlying explanation is that MMSE and IRM, when conducted

on the training set, result in an over-smooth trajectory for speech utterances in the time or spectral domains and therefore reduce the diversity and discriminant capability of the various phone realizations in the training set.

3.  As for the second arrangement (i.e., MMSE and IRM conducted only on the test set), IRM appears to bring moderately lower WERs for the two noise types "white" and "engine" (shown in the lower part of Tables 1 and 2), while it is not quite helpful for the "jackhammer" case (shown in the lower part of Table 3). Comparatively, MMSE worsens the recognition accuracy for all three noise situations.

Based on the above observations, we may conclude with the following points:

1.  The two metrics, PESQ and WER, seem to have little correlation since the method MMSE, which causes better PESQ scores, results in higher WERs (except for some low SNR cases). That is, improving the speech quality does not necessarily reduce its recognition accuracy. Relative to PESQ, the STOI index has more to do with the recognition accuracy in more noisy situations. This can be observed in that the method IRM can improve STOI and recognition accuracy simultaneously for "white" and "engine" noises, even though the improvement is somewhat marginal. In addition, the increase in STOI does not necessarily reduce WER, while the reduction in WER always comes with the increase in STOI.

2.  The two SE methods used here do not show obvious benefits of speech recognition in noisy situations. In our opinion, directly compensating speech features used for a noisy recognition system is probably more helpful in reducing WERs than enhancing speech signals by SE methods.

**Table 1.** The various evaluation results of MMSE, IRM, and the baseline for the test set in noise environment "**white**".The results worse than the baseline are marked with a superscript "*". In particular, we also give the same marking for the results worse than the baseline in the other tables.

|  |  | Signal-To-Noise Ratio (SNR) | | | | | |
|---|---|---|---|---|---|---|---|
|  |  | **−6 dB** | **−3 dB** | **0 dB** | **3 dB** | **6 dB** | **12 dB** |
| PESQ | baseline | 1.031 | 1.036 | 1.044 | 1.062 | 1.096 | 1.271 |
|  | MMSE | 1.040 | 1.054 | 1.080 | 1.130 | 1.237 | 1.647 |
|  | IRM | 1.032 | 1.036 | 1.044 | 1.063 | 1.098 | 1.275 |
| STOI | baseline | 0.577 | 0.648 | 0.722 | 0.790 | 0.850 | 0.935 |
|  | MMSE | 0.584 | 0.643 * | 0.697 * | 0.734 * | 0.766 * | 0.840 * |
|  | IRM | 0.578 | 0.649 | 0.723 | 0.792 | 0.852 | 0.937 |
| WER (%) | baseline | 66.1 | 62.1 | 57.0 | 49.8 | 44.8 | 34.4 |
|  | MMSE (TR, TS) | 70.0 * | 66.7 * | 61.9 * | 56.5 * | 50.5 * | 40.5 * |
|  | IRM (TR, TS) | 67.3 * | 63.1 * | 58.3 * | 52.5 * | 47.3 * | 37.5 * |
|  | MMSE (TS) | 70.3 * | 66.8 * | 60.4 * | 55.0 * | 49.7 * | 40.1 * |
|  | IRM (TS) | 65.8 | 61.9 | 56.5 | 50.4 | 44.4 | 34.3 |

**Table 2.** The various evaluation results of MMSE, IRM, and the baseline for the test set in noise environment "**engine**".

|  |  | Signal-To-Noise Ratio (SNR) | | | | | |
|---|---|---|---|---|---|---|---|
|  |  | −6 dB | −3 dB | 0 dB | 3 dB | 6 dB | 12 dB |
| PESQ | baseline | 1.051 | 1.064 | 1.087 | 1.128 | 1.195 | 1.482 |
|  | MMSE | 1.103 | 1.103 | 1.198 | 1.350 | 1.520 | 1.967 |
|  | IRM | 1.051 | 1.064 | 1.087 | 1.128 | 1.195 | 1.493 |
| STOI | baseline | 0.506 | 0.575 | 0.651 | 0.726 | 0.796 | 0.899 |
|  | MMSE | 0.515 | 0.579 | 0.638 * | 0.692 * | 0.743 * | 0.850 * |
|  | IRM | 0.506 | 0.575 | 0.652 | 0.726 | 0.796 | 0.901 |
| WER (%) | baseline | 65.3 | 61.7 | 55.1 | 48.2 | 41.5 | 31.1 |
|  | MMSE (TR,TS) | 68.2 * | 63.8 * | 57.8 * | 51.4 * | 44.3 * | 32.6 * |
|  | IRM (TR,TS) | 67.5 * | 64.7 * | 59.0 * | 52.9 * | 46.0 * | 35.3 * |
|  | MMSE (TS) | 68.3 * | 63.6 * | 57.3 * | 51.4 * | 44.9 * | 34.2 * |
|  | IRM (TS) | 65.9 * | 61.4 | 54.8 | 48.2 | 41.2 | 31.1 |

**Table 3.** The various evaluation results of MMSE, IRM, and the baseline for the test set in noise environment "**jackhammer**".

|  |  | Signal-To-Noise Ratio (SNR) | | | | | |
|---|---|---|---|---|---|---|---|
|  |  | −6 dB | −3 dB | 0 dB | 3 dB | 6 dB | 12 dB |
| PESQ | baseline | 1.682 | 1.946 | 2.242 | 2.566 | 2.919 | 3.533 |
|  | MMSE | 2.087 | 2.413 | 2.739 | 3.059 | 3.323 | 3.812 |
|  | IRM | 1.773 | 2.044 | 2.350 | 2.680 | 3.024 | 3.624 |
| STOI | baseline | 0.924 | 0.950 | 0.968 | 0.981 | 0.989 | 0.997 |
|  | MMSE | 0.917 * | 0.944 * | 0.962 * | 0.976 * | 0.983 * | 0.991 * |
|  | IRM | 0.932 | 0.955 | 0.970 | 0.982 | 0.990 | 0.997 |
| WER (%) | baseline | 31.4 | 27.8 | 25.9 | 23.8 | 23.0 | 21.9 |
|  | MMSE (TR,TS) | 32.3 * | 28.7 * | 26.4 * | 24.5 * | 23.9 * | 22.5 * |
|  | IRM (TR,TS) | 33.6 * | 30.4 * | 27.4 * | 25.2 * | 24.1 * | 22.9 * |
|  | MMSE (TS) | 32.9 * | 29.0 * | 27.2 * | 25.2 * | 24.4 * | 23.2 * |
|  | IRM (TS) | 32.4 * | 29.8 * | 26.6 * | 25.2 * | 23.8 * | 22.8 * |

*4.3. Experimental Results and Discussions for the Proposed Method*

In this subsection, we provide the experimental results of our presented method and some discussions. For ease of discussion, the presented method is termed "maximum state posterior probability" with the abbreviation "MSPP". The experimental results for the multi-condition test utterances are further split into two parts: multi-condition training mode and clean-condition training mode. Note that the denoising framework in our method is with the help of the multi-condition training set, but we would like to see if the resulting enhanced speech features can behave well in both modes.

Furthermore, we employ a comparative method for evaluation which primarily uses a deep neural network (DNN) to update the input FBANK features. This DNN is learned by directly minimizing the mean squared error (MSE) between the noisy and clean FBANK

feature pairs in the multi-condition training set. This comparative method is termed feature-based MSE and abbreviated by "FMSE" in the following discussions.

Before going into the experimental results, it is worth noting again that we used the original plain FBANK in the training set to prepare the acoustic models for both multi-condition and clean-condition training modes. Therefore, the various speech enhancement/robust feature extraction methods were just conducted on the test sets.

### 4.3.1. Multi-Condition Training Mode

Tables 4–6 list the WER (%) for our method, MSPP, and FMSE, together with the two SE methods (MMSE and IRM) in the preceding sub-section **using the acoustic models learned from the multi-condition training set**. The used acoustic models are common to the four comparative methods. Since the presented MSPP is directly related to state posteriors in the acoustic models, here we also list the state error rates with the various methods for the test utterances. From these three tables, we have the following observations:

1. On average, Table 6 shows that the WERs achieved by various methods in the jackhammer noise environment are significantly lower than those in the white and engine noise environments as in Tables 4 and 5, indicating that jackhammer noise brings less distortion to speech signals compared with the white and engine noises. However, we find that all of the used methods here fail to outperform the baseline results for the jackhammer noise case, indicating that the speech enhancement/noise-robust feature methods might introduce further observable distortion to less contaminated utterances.

2. As for the white and engine noise situations as in Tables 4 and 5, the newly proposed MSPP can achieve lower WERs in most SNR cases and outperform the other methods in comparison, which clearly validates the main thought of MSPP that increasing the state posteriors helps to improve the noise robustness and promote recognition accuracy. For example, observing the the lower part of Tables 4 and 5, MSPP achieves 54.9% and 54.6% WER at 0 dB SNR for white and engine noise cases, better than the baseline results 57.0% and 55.1%. In particular, the denoising framework in the presented MSPP is created upon the noisy dataset in which noise types are neither white nor engine. Thus, MSPP is shown to have a generalization capability to conquer unseen noise.

3. When observing the state error rates achieved by the various methods as shown in the upper parts of Tables 4–6, we do not see they strongly correlate with the obtained WERs. For example, the presented MSPP does not always have a lower state error rate, while the method IRM, which has the lowest state error rate in some SNR cases, results in higher WERs. One possible explanation is that the state error rate is quite sensitive to distortion, making it not feasible to be a good evaluation metric. In addition, the presented MSPP attempts to choose the best possible state sequence and is a decision process, which does not necessarily coincide with the minimization of the state error, an estimation process.

4. The method FMSE, which is developed to minimize the mean squared error (MSE) between the clean–noisy FBANK pairs, behaves worse than the baseline in almost all noise situations. For example, in Tables 4 and 5, FMSE achieves 57.2% and 56.2% WER at 0 dB SNR for white and engine noise cases, worse than the baseline results 57.0% and 55.1%. These results seem to agree with our earlier discussions in Section 1, i.e., the mismatch in evaluation (optimization) metrics makes the MMSE-wise transformed speech features give rise to worse recognition accuracy. Another reason is that the learned DNN in FMSE overfits the training data and thus fails to enhance the test data well.

**Table 4.** The state error rate and WER (%) of the baseline, MSPP, FMSE, MMSE, and IRM with multi-condition training for the test set in noise environment "**white**".

| | | Signal-To-Noise Ratio (SNR) | | | | | |
|---|---|---|---|---|---|---|---|
| | | −6 dB | −3 dB | 0 dB | 3 dB | 6 dB | 12 dB |
| state error rate | baseline | 0.951 | 0.928 | 0.895 | 0.849 | 0.807 | 0.717 |
| | MSPP | 0.949 | 0.925 | 0.895 | 0.852 * | 0.809 * | 0.723 * |
| | FMSE | 0.954 * | 0.930 * | 0.899 * | 0.856 * | 0.811 * | 0.729 * |
| | MMSE | 0.968 * | 0.952 * | 0.929 * | 0.899 * | 0.867 * | 0.794 * |
| | IRM | 0.950 | 0.925 | 0.891 | 0.850 * | 0.803 | 0.718 * |
| WER (%) | baseline | 66.1 | 62.1 | 57.0 | 49.8 | 44.8 | 34.4 |
| | MSPP | 65.5 | 61.0 | 54.9 | 48.9 | 43.2 | 34.7 * |
| | FMSE | 69.2 * | 63.3 * | 57.2 * | 50.4 * | 44.8 | 35.3 * |
| | MMSE | 70.3 * | 66.8 * | 60.4 * | 55.0 * | 49.7 * | 40.1 * |
| | IRM | 65.8 | 61.9 | 56.5 | 50.4 * | 44.4 | 34.3 |

**Table 5.** The state error rate and WER (%) of the baseline, MSPP, FMSE, MMSE, and IRM with multi-condition training for the test set in noise environment "**engine**".

| | | Signal-To-Noise Ratio (SNR) | | | | | |
|---|---|---|---|---|---|---|---|
| | | −6 dB | −3 dB | 0 dB | 3 dB | 6 dB | 12 dB |
| state error rate | baseline | 0.958 | 0.936 | 0.898 | 0.850 | 0.796 | 0.689 |
| | MSPP | 0.957 | 0.933 | 0.899 * | 0.851 * | 0.799 * | 0.694 * |
| | FMSE | 0.956 | 0.932 | 0.892 | 0.840 | 0.782 | 0.676 |
| | MMSE | 0.964 * | 0.946 * | 0.917 * | 0.878 * | 0.832 * | 0.730 * |
| | IRM | 0.958 | 0.936 | 0.898 | 0.851 * | 0.795 | 0.690 * |
| WER (%) | baseline | 65.3 | 61.7 | 55.1 | 48.2 | 41.5 | 31.1 |
| | MSPP | 65.5 * | 60.2 | 54.6 | 47.9 | 41.7 * | 32.3 * |
| | FMSE | 70.4 * | 64.5 * | 56.2 * | 48.2 | 41.0 | 31.2 * |
| | MMSE | 68.3 * | 63.6 * | 57.3 * | 51.4 * | 44.9 * | 34.2 * |
| | IRM | 65.9 * | 61.4 | 54.8 | 48.2 | 41.2 | 31.1 |

**Table 6.** The state error rate and WER (%) of the baseline, MSPP, FMSE, MMSE, and IRM with multi-condition training for the test set in noise environment "**jackhammer**".

| | | Signal-To-Noise Ratio (SNR) | | | | | |
|---|---|---|---|---|---|---|---|
| | | −6 dB | −3 dB | 0 dB | 3 dB | 6 dB | 12 dB |
| state error rate | baseline | 0.693 | 0.656 | 0.622 | 0.600 | 0.585 | 0.568 |
| | MSPP | 0.704 * | 0.667 * | 0.638 * | 0.614 * | 0.598 * | 0.579 * |
| | FMSE | 0.725 * | 0.692 * | 0.663 * | 0.644 * | 0.630 * | 0.618 * |
| | MMSE | 0.712 * | 0.678 * | 0.647 * | 0.625 * | 0.610 * | 0.591 * |
| | IRM | 0.707 * | 0.675 * | 0.645 * | 0.618 * | 0.604 * | 0.585 * |

**Table 6.** *Cont.*

|  |  | Signal-To-Noise Ratio (SNR) | | | | | |
|---|---|---|---|---|---|---|---|
|  |  | −6 dB | −3 dB | 0 dB | 3 dB | 6 dB | 12 dB |
| WER (%) | baseline | 31.4 | 27.8 | 25.9 | 23.8 | 23.0 | 21.9 |
|  | MSPP | 32.6 * | 29.4 * | 27.5 * | 25.7 * | 25.1 * | 23.9 * |
|  | FMSE | 34.4 * | 31.3 * | 29.1 * | 27.6 * | 27.0 * | 26.1 * |
|  | MMSE | 32.9 * | 29.0 * | 27.2 * | 25.2 * | 24.4 * | 23.2 * |
|  | IRM | 32.4 * | 29.8 * | 26.6 * | 25.2 * | 23.8 * | 22.8 * |

### 4.3.2. Clean-Condition Training Mode

In Tables 7–9, we present the WER results of the presented MSPP, FMSE, and two SE methods, MMSE and IRM, for the test set under the clean-condition training scenario. Please note that here we use the **original** FBANK features in the clean training set to train the acoustic models, which are common to those evaluation methods. Several observations can be found in these tables:

1. Compared with the multi-condition baseline results shown in the previous three tables (Tables 4–6), the clean-condition baseline results look worse. For example, at 0 dB SNR for the white noise case, the clean-condition baseline WER is 61.0% (in Table 7), worse than the multi-condition baseline WER 57.0% (in Table 4). This is probably because the clean-condition training data are more mismatched with the test set than the multi-condition training data.

**Table 7.** The state error rate and WER (%) of the baseline, MSPP, FMSE, MMSE, and IRM with clean-condition training for the test set in noise environment "**white**".

|  |  | Signal-To-Noise Ratio (SNR) | | | | | |
|---|---|---|---|---|---|---|---|
|  |  | −6 dB | −3 dB | 0 dB | 3 dB | 6 dB | 12 dB |
| state error rate | baseline | 0.968 | 0.953 | 0.928 | 0.898 | 0.864 | 0.787 |
|  | MSPP | 0.961 | 0.942 | 0.914 | 0.881 | 0.841 | 0.757 |
|  | FMSE | 0.962 | 0.943 | 0.919 | 0.886 | 0.851 | 0.767 |
|  | MMSE | 0.977 * | 0.966 * | 0.948 * | 0.927 * | 0.899 * | 0.836 * |
|  | IRM | 0.968 | 0.950 | 0.926 | 0.896 | 0.863 | 0.788 * |
| WER (%) | baseline | 67.6 | 64.6 | 61.0 | 55.6 | 50.9 | 41.2 |
|  | MSPP | 64.3 | 60.6 | 56.3 | 50.8 | 45.4 | 36.6 |
|  | FMSE | 68.6 * | 64.3 | 58.9 | 53.0 | 47.7 | 38.9 |
|  | MMSE | 69.9 * | 67.1 * | 63.7 * | 59.1 * | 54.4 * | 45.6 * |
|  | IRM | 67.4 | 64.3 | 60.6 | 55.0 | 50.5 | 40.7 |

2. The two SE methods, MMSE and IRM, give similar or higher WERs compared to the baseline results. For example, at 0 dB SNR for the engine noise case (in Table 8), the WERs for the baseline, MMSE, and IRM are 61.1%, 62.5%, and 60.6%, respectively. It again shows that enhancing the utterances does not necessarily improve the respective recognition accuracy.

3. The presented MSPP is shown to provide significantly lower WERs than the baseline results for most noisy cases (except for the jackhammer noise at the SNRs higher than −3 dB). For example, at 0 dB SNR for the white noise case (in Table 7), the WERs for the baseline and MSPP are 61.0% and 56.3%, respectively. These results indicate

that MSPP promises to promote the recognition accuracy of noisy speech features even when MSPP does not pre-process the training set for the acoustic model. It reconfirms our previous statement: MSPP has a generalization capability to conquer unseen noise.

**Table 8.** The state error rate and WER (%) of the baseline, MSPP, FMSE, MMSE, and IRM with clean-condition training for the test set in noise environment "**engine**".

| | | Signal-To-Noise Ratio (SNR) | | | | | |
|---|---|---|---|---|---|---|---|
| | | −6 dB | −3 dB | 0 dB | 3 dB | 6 dB | 12 dB |
| state error rate | baseline | 0.973 | 0.959 | 0.938 | 0.909 | 0.870 | 0.790 |
| | MSPP | 0.967 | 0.948 | 0.919 | 0.881 | 0.832 | 0.732 |
| | FMSE | 0.966 | 0.946 | 0.919 | 0.882 | 0.832 | 0.734 |
| | MMSE | 0.977 * | 0.965 * | 0.947 * | 0.917 * | 0.882 * | 0.799 * |
| | IRM | 0.974 * | 0.958 | 0.937 | 0.906 | 0.870 | 0.786 |
| WER (%) | baseline | 67.3 | 64.7 | 61.1 | 56.3 | 50.4 | 39.4 |
| | MSPP | 64.4 | 60.9 | 55.2 | 49.8 | 43.8 | 34.7 |
| | FMSE | 69.9 * | 65.5 * | 59.7 | 52.3 | 46.8 | 36.6 |
| | MMSE | 69.1 * | 65.9 * | 62.5 * | 57.3 * | 52.1 * | 40.7 * |
| | IRM | 66.8 | 65.1 * | 60.6 | 55.7 | 49.7 | 39.4 |

**Table 9.** The state error rate and WER (%) of the baseline, MSPP, FMSE, MMSE, and IRM with clean-condition training for the test set in noise environment "**jackhammer**".

| | | Signal-To-Noise Ratio (SNR) | | | | | |
|---|---|---|---|---|---|---|---|
| | | −6 dB | −3 dB | 0 dB | 3 dB | 6 dB | 12 dB |
| state error rate | baseline | 0.739 | 0.701 | 0.677 | 0.638 | 0.612 | 0.576 |
| | MSPP | 0.720 | 0.682 | 0.653 | 0.625 | 0.606 | 0.583 * |
| | FMSE | 0.733 | 0.692 | 0.660 | 0.633 | 0.614 * | 0.587 * |
| | MMSE | 0.746 * | 0.710 * | 0.676 | 0.645 * | 0.621 * | 0.585 * |
| | IRM | 0.728 | 0.692 | 0.660 | 0.631 | 0.610 | 0.575 |
| WER (%) | baseline | 35.3 | 31.5 | 28.5 | 26.7 | 24.9 | 23.1 |
| | MSPP | 33.8 | 30.6 | 28.7 * | 27.0 * | 26.3 * | 24.9 * |
| | FMSE | 35.0 | 32.2 * | 29.7 * | 28.1 * | 27.0 * | 25.4 * |
| | MMSE | 36.5 * | 32.8 * | 29.6 * | 27.7 * | 25.3 * | 23.7 * |
| | IRM | 34.8 | 31.8 * | 28.8 * | 26.6 | 24.9 | 23.3 * |

*4.4. Experimental Results and Discussions for the Proposed Method with Data Augmentation*

In this subsection, we provide the experimental results of a variant of our presented method, which further adopted the data augmentation technique [40]. To realize it, the DNN-HMM acoustic models were retrained with the original FBANKs and MSPP-enhanced FBANKs in the multi-condition training set (the original acoustic models were trained with the original FBANKs only). In particular, the training set here was arranged such that 50% of the utterances were converted to plain FBANKs while the remaining 50% corresponded to MSPP-enhanced FBANKs. The resulting acoustic models were evaluated on MSPP-enhanced FBANKs in the noisy test set to see how the recognition accuracy would be influenced.

In Tables 10–12, we present the state error rates and WER (%) under the multi-condition training scenario with respect to the aforementioned two acoustic model configurations. Here, the configuration using data augmentation is termed "MSPP-Aug". From the three tables, we see that with data augmentation, the resulting acoustic models can further reduce the WERs and the state error rates for the MSPP-enhanced test set compared with the original FBANK-wise acoustic models. The WER reduction can be as high as 1% for some SNR cases for the noise types "white" and "engine". For example, at 0 dB SNR for the engine noise case (in Table 11), the WERs for the baseline, MSPP, and MSPP-Aug are 55.1%, 54.6%, and 53.0%, respectively. These results clearly reveal the effectiveness of the data augmentation technique, which increases the diversity of training data and thus benefits the noise robustness of the resulting acoustic models.

**Table 10.** The state error rate and WER (%) of the baseline, MSPP, and MSPP-Aug with multi-condition training for the test set in noise environment "**white**".

|  |  | Signal-To-Noise Ratio (SNR) | | | | | |
|---|---|---|---|---|---|---|---|
|  |  | −6 dB | −3 dB | 0 dB | 3 dB | 6 dB | 12 dB |
| state error rate | baseline | 0.951 | 0.928 | 0.895 | 0.849 | 0.807 | 0.717 |
|  | MSPP | 0.949 | 0.925 | 0.895 | 0.852 * | 0.809 * | 0.723 * |
|  | MSPP-Aug | 0.948 | 0.923 | 0.892 | 0.851 * | 0.808 * | 0.720 * |
| WER (%) | baseline | 66.1 | 62.1 | 57.0 | 49.8 | 44.8 | 34.4 |
|  | MSPP | 65.5 | 61.0 | 54.9 | 48.9 | 43.2 | 34.7 * |
|  | MSPP-Aug | 64.3 | 59.9 | 54.0 | 48.4 | 42.7 | 34.0 |

**Table 11.** The state error rate and WER (%) of the baseline, MSPP, and MSPP-Aug with multi-condition training for the test set in noise environment "**engine**".

|  |  | Signal-To-Noise Ratio (SNR) | | | | | |
|---|---|---|---|---|---|---|---|
|  |  | −6 dB | −3 dB | 0 dB | 3 dB | 6 dB | 12 dB |
| state error rate | baseline | 0.958 | 0.936 | 0.898 | 0.850 | 0.796 | 0.689 |
|  | MSPP | 0.957 | 0.933 | 0.899 * | 0.851 * | 0.799 * | 0.694 * |
|  | MSPP-Aug | 0.955 | 0.930 | 0.894 | 0.846 | 0.793 | 0.690 * |
| WER (%) | baseline | 65.3 | 61.7 | 55.1 | 48.2 | 41.5 | 31.1 |
|  | MSPP | 65.5 * | 60.2 | 54.6 | 47.9 | 41.7 * | 32.3 * |
|  | MSPP-Aug | 64.2 | 60.0 | 53.0 | 47.0 | 41.1 | 31.8 * |

**Table 12.** The state error rate and WER (%) of the baseline, MSPP, and MSPP-Aug with multi-condition training for the test set in noise environment "**jackhammer**".

|  |  | Signal-To-Noise Ratio (SNR) | | | | | |
|---|---|---|---|---|---|---|---|
|  |  | −6 dB | −3 dB | 0 dB | 3 dB | 6 dB | 12 dB |
| state error rate | baseline | 0.693 | 0.656 | 0.622 | 0.600 | 0.585 | 0.568 |
|  | MSPP | 0.704 * | 0.667 * | 0.638 * | 0.614 * | 0.598 * | 0.579 * |
|  | MSPP-Aug | 0.694 * | 0.659 * | 0.627 * | 0.608 * | 0.595 * | 0.578 * |
| WER (%) | baseline | 31.4 | 27.8 | 25.9 | 23.8 | 23.0 | 21.9 |
|  | MSPP | 32.6 * | 29.4 * | 27.5 * | 25.7 * | 25.1 * | 23.9 * |
|  | MSPP-Aug | 31.9 * | 29.4 * | 26.9 * | 25.2 * | 24.2 * | 23.1 * |

## 5. Conclusions and Future Work

In this study, we primarily pay attention to the noise issue in automatic speech recognition (ASR). We analyze the possible effects of speech enhancement methods on acoustic feature robustness in ASR and then present a novel deep-learning-based framework to create noise-robust speech features. This framework exploits a deep neural network to maximize the state posteriors of the used acoustic models which provided the clean–noisy FBANK speech feature pairs in the training set. The preliminary experiments reveal that the newly presented method can improve the recognition accuracy of FBANK features, particularly under the moderate and severe noise situations. It can behave well regardless of the training set being mismatched multi-condition or clean noise-free condition that prepares the acoustic models. As for future avenues, we could further enhance the presented denoising network by adopting more versatile training data or increasing the amount of training data. We also plan to integrate this method with some other feature-based or model-based noise-robust algorithms to achieve even better performance.

**Author Contributions:** Conceptualization, L.-C.C. and J.-W.H.; methodology, L.-C.C. and J.-W.H.; software, L.-C.C.; validation, L.-C.C. and J.-W.H.; formal analysis, L.-C.C. and J.-W.H.; investigation, L.-C.C. and J.-W.H.; resources, J.-W.H.; data curation, L.-C.C. and J.-W.H.; writing—original draft preparation, L.-C.C. and J.-W.H.; writing—review and editing, J.-W.H.; visualization, L.-C.C. and J.-W.H.; supervision, J.-W.H.; project administration, J.-W.H.; funding acquisition, J.-W.H. All authors have read and agreed to the published version of the manuscript.

**Funding:** This research was funded by National Science Council (NSC) in Taiwan with grant number 110-2221-E-260-012.

**Data Availability Statement:** Not applicable.

**Conflicts of Interest:** The authors declare no conflict of interest.

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
