# Peer review of "A Preliminary Study of Robust Speech Feature Extraction Based on Maximizing the Probability of States in Deep Acoustic Models"

_asi, doi:10.3390/asi5040071_

Round 1
Reviewer 1 Report
This paper presents a novel deep learning-based noise-robust speech feature extraction algorithm using a loss function directly associated with the performance of a speech recognition system in noisy environments. An argument is presented for ignoring MSE as a loss function for training DNNs for speech enhancement. The overall intent of the work is well motivated and the discussion of results illuminating though not clear cut.
Overall the paper needs further proofreading for minor mechanical issues (capitalization and verb tense) as well as continuity (e.g. ASR as an acronym is used throughout but only defined in section 5). Reformatting would also help as presenting tables of results interspersed with references is quite odd.
The flowchart of the proposed method in figure 2 is not helpful in in its current form, a larger flow chart of process steps would be appropriate in its place.
The discussion of results is helpful, but could be better if the individual points were more directly tied to the particular tables of results. An interested reader can find support for the authors' points in the tables, but including more specific call outs to particular table cells would improve the discussion.
Author Response
We are very grateful to the reviewers for their patience in carefully reading this submission and providing many constructive/valuable suggestions. Below are the responses to the major comments raised by the reviewers.
(1) This paper presents a novel deep learning-based noise-robust speech feature extraction algorithm using a loss function directly associated with the performance of a speech recognition system in noisy environments. An argument is presented for ignoring MSE as a loss function for training DNNs for speech enhancement. The overall intent of the work is well motivated and the discussion of results illuminating though not clear cut.
Response: Thank you very much for this comment. We have enriched the section of experimental results and discussion by giving examples shown in the tables of experimental results. The corresponding revisions are highlighted in pages 10-17 of the file ``Track Changes.pdf’’. (Please see the attachment)
(2) Overall the paper needs further proofreading for minor mechanical issues (capitalization and verb tense) as well as continuity (e.g. ASR as an acronym is used throughout but only defined in section 5). Reformatting would also help as presenting tables of results interspersed with references is quite odd.
Response: Thank you very much for this comment. We have corrected the typo and grammatical issues as well as acronym definition and discussion by giving examples shown in the tables of experimental results. The corresponding revisions are highlighted in the file ``Track Changes.pdf’’. (Please see the attachment)
(3) The flowchart of the proposed method in figure 2 is not helpful in in its current form, a larger flow chart of process steps would be appropriate in its place.
Response: Thank you very much for this comment. We have modified the flowchart of the proposed method as a new figure (Figure 2), which is in page 7 of the file ``Track Changes.pdf’’. (Please see the attachment)
(4) The discussion of results is helpful, but could be better if the individual points were more directly tied to the particular tables of results. An interested reader can find support for the authors' points in the tables, but including more specific call outs to particular table cells would improve the discussion.
Response: Thank you very much for this comment. We have enriched the section of experimental results by tying the individual points to some experimental results shown in the specified tables. The corresponding revisions are highlighted in pages 10-17 of the file ``Track Changes.pdf’’. (Please see the attachment)

Reviewer 2 Report
1) For proper understanding of article, author must include detailed review of related article and must clear research gap.
2) Author must compare his results from latest paper.
Author Response
We are very grateful to the reviewers for their patience in carefully reading this submission and providing many constructive/valuable suggestions. Below are the responses to the major comments raised by the reviewers.
(1) For proper understanding of article, author must include detailed review of related article and must clear research gap.
Response: Thank you very much for this comment. We have added a paragraph to review some state-of-the-art noise robust automatic speech recognition (ASR) techniques in section 1. The corresponding contents are highlighted in pages 2-3 of the file ``Track Changes.pdf’’. (Please see the attachment.)
(2) Author must compare his results from latest paper.
Response: We have had some comments about the presented method and claimed that it can be well integrated with some latest advanced methods to boost the ASR performance. Due to limited time and computing resources, we does not implement these latest advanced methods to do the comparison. The corresponding comments are ``Compared with these complicated and fine-grinned techniques, our newly presented method is a relatively lightweight network that is much easier to be learned, while it is likely less effective. However, our method is a DNN-wise transformation in the same feature domain and thus can be easily integrated with these advanced methods to boost the ASR performance.’’
These revisions are highlighted in page 3 of the file ``Track Changes.pdf’’. (Please see the attachment.)
